# Influence of Iron on the Gut Microbiota in Colorectal Cancer

**DOI:** 10.3390/nu12092512

**Published:** 2020-08-20

**Authors:** Oliver Phipps, Hafid O. Al-Hassi, Mohammed N. Quraishi, Aditi Kumar, Matthew J. Brookes

**Affiliations:** 1Research Institute in Healthcare Science, Faculty of Science and Engineering, University of Wolverhampton, Wolverhampton WV1 1LY, UK; h.omar6@wlv.ac.uk (H.O.A.-H.); aditikumar@nhs.net (A.K.); matthew.brookes@nhs.net (M.J.B.); 2Institute of Cancer and Genomic Sciences, University of Birmingham, Birmingham B15 2TT, UK; m.n.quraishi@bham.ac.uk; 3Royal Wolverhampton Hospitals NHS Trust, Gastroenterology Unit, Wolverhampton WV10 0QP, UK

**Keywords:** iron, microbiota, protective, pathogenic, bacteria, colorectal cancer, iron therapy

## Abstract

Perturbations of the colonic microbiota can contribute to the initiation and progression of colorectal cancer, leading to an increase in pathogenic bacteria at the expense of protective bacteria. This can contribute to disease through increasing carcinogenic metabolite/toxin production, inducing inflammation, and activating oncogenic signaling. To limit disease progression, external factors that may influence the colonic microbiota need to be considered in patients with colorectal cancer. One major factor that can influence the colonic microbiota is iron. Iron is an essential micronutrient that is required by both prokaryotes and eukaryotes for cellular function. Most pathogenic bacteria have heightened iron acquisition mechanisms and therefore tend to outcompete protective bacteria for free iron. Colorectal cancer patients often present with anemia due to iron deficiency, and thus they require iron therapy. Depending upon the route of administration, iron therapy has the potential to contribute to a procarciongenic microbiota. Orally administered iron is the common treatment for anemia in these patients but can lead to an increased gut iron concentration. This suggests the need to reassess the route of iron therapy in these patients. Currently, this has only been assessed in murine studies, with human trials being necessary to unravel the potential microbial outcomes of iron therapy.

## 1. Introduction

The gut microbiota comprises an intricate population of microorganisms, which exert a complex relationship with their host during both homeostasis and disease. Numerous gastrointestinal diseases, including colorectal cancer, are associated with changes in gut microbiota compared to healthy controls [1]. These perturbations, often referred to as dysbiosis, may have a role in contributing to disease mechanisms associated with colitis, tumor initiation, and tumor progression [2,3]. This dysregulation in the gut microbiota is potentially associated with a shift away from strains of bacteria beneficial to the host (symbionts), which act to limit the development of such diseases, substituting with potentially pathogenic bacteria (pathobionts) that can contribute to inflammation and carcinogenic metabolite production [4]. As microbial alterations are pre-existent in colorectal cancer, special care needs to be considered in order to limit external influences that may further shift microbial populations towards a procarciongenic microbiota [5]. One critical factor that needs to be considered is iron, as this nutrient is required by many bacterial cells to grow and survive [6]. Patients with colorectal cancer often present with anemia due to iron deficiency and are subsequently treated with iron supplements. If enteral iron is given, this can contribute to an increased luminal iron pool within the gut [7,8]. Many pathogenic bacteria have high-affinity iron acquisition mechanisms such as siderophores and transferrin/lactoferrin receptors to obtain free iron from their environment. Therefore, excessive luminal iron may select for pathogenic bacteria at the expense of beneficial commensal bacteria [9,10]. The role of increased iron in promoting pathogenic bacteria has been evaluated in an in vitro study by Kortman et al., who assessed the growth of prevalent enteric pathogens in high and low iron cultures. Growth of pathogenic bacteria such as *Salmonella typhimurium* was increased in response to high iron and showed increased adhesion and invasion of colorectal cancer cells [11]. This supports the idea that oral iron supplementation can promote the growth and potentially the virulence of pathogenic bacteria through increasing colonic iron availability [9,11]. Increased colonic iron concentration through iron supplementation has also been shown to promote pathogenic bacterial species in iron-deficient Kenyan infants while decreasing protective bacteria such as *Bifidobacteria* and *lactobacillus*. This supports the role of increased luminal iron promoting pathogenic bacteria at the expense of beneficial bacteria, however, the underlying mechanisms are not fully understood [12,13]. The relationship between the microbiota and colorectal cancer will be discussed in this review, focusing on how iron can alter bacterial populations and the outcomes of iron supplementation. 

## 2. Microbiota in Colonic Homeostasis and Disease

### 2.1. Gut Microbiota

The human intestine provides a platform for a complex and dynamic ecosystem, consisting of greater than 10^13^ microorganisms. This collection of microorganisms residing within the gastrointestinal tract consists of bacteria, viruses, and fungi and is collectively termed the gut microbiota [14]. The gut microbiota consists of over 400 bacterial species that colonize the lumen and mucosa of the colon but do not penetrate a healthy bowel wall [3]. The predominant phyla residing within the gut are Firmicutes *(≈*49–76%*),* Bacteroidetes (*≈*16–23%), Proteobacteria (<10%) and Actinobacteria (<5%), with a large diversity at the lower taxonomic levels [9].

Bacterial colonization of the colon can be either commensal or pathogenic [15]. Commensal bacteria can either derive benefit from the host, but the host receives no benefit or harm from the bacteria, or commensal bacteria can provide a mutualistic relationship with the host when both bacteria and host benefit. This can act to maintain host homeostasis through contributing to fiber digestion, synthesis of vitamins, and enterohepatic deconjugation of metabolites [16]. These commensal bacteria also contribute host defense against colonic invasion of pathogens, due to both requiring an analogous ecological niche. Hence, under normal physiological conditions, the commensal bacteria act to outcompete potential pathogenic colonization of the colon, therefore preventing infection [17]. The gut microbiota generally have a commensal relationship with their host. Nevertheless, there is the potential for this relationship to be disrupted through an alteration in the composition of bacteria. This occurs through an imbalance in bacterial populations through the replacement of non-harmful commensal bacteria with pathogenic bacteria. These pathogenic bacteria can cause harm to the host through invading the bowel wall, causing inflammation and inducing carcinogenic signaling and metabolites [18]. Therefore, these bacteria have the potential to contribute to tumorigenesis in the colon; however, what remains unclear is whether the presence of these bacteria are the cause or consequence of colorectal cancer [2]. Many studies, as discussed in this review, support the role of the microbiota in the development of colorectal cancer. However, tumors themselves have the ability to shape the composition of the microbiota, with tumor genetic profiles being able to differentially influence the microbiota [19]. This has been assessed in colorectal tumors in a study by Burns et al., showing that prevalent loss of function (LoF) mutations in cancer-related genes, including *APC*, *ANKRD36C*, *CTBP2*, *KMT2C*, and *ZNF71*, are correlated with defined bacterial populations. The LoF mutations were linked with microbial profiles that were associated with cancer pathways, including mitogen-activated protein (MAP) kinase and Wnt signaling [20]. This suggests that the tumor has the ability to regulate the microbiota, through cancer genetics, in order to potentially support tumor progression [19,20].

### 2.2. Bacteria and Cancer

The correlation between bacterial infection and cancer is predominantly through three mechanisms: initiating chronic inflammation, producing carcinogenic metabolites and toxins, and activating oncogenic signaling (Figure 1) [21]. The most prominent association between bacteria and cancer is *Helicobacter pylori* and gastric cancer [22]. *Helicobacter pylori* is a Gram-negative bacterium that colonizes the lining of the stomach. Here, it may induce chronic gastritis, which can contribute to the development of gastric ulcers and stomach cancer [23,24]. *Helicobacter pylori* also produce virulence factors such as cytotoxin-associated gene A and vacuolating cytotoxin A, which can contribute to the damage and malignant transformation of the gastric epithelia [25]. It is estimated that *Helicobacter pylori-*positive patients have a 10–20% lifetime risk of developing peptic ulcers and a 1–2% risk of developing gastric cancer [26]. *Helicobacter pylori* have also been associated with other pathologies such as colorectal polyps and cancer [27]. 

The association between *Helicobacter pylori* and gastric cancer has been confirmed through fulfilling Koch’s postulates, as *Helicobacter pylori* infection can lead to gastric cancer in a susceptible host [28,29]. The same association has not been confirmed between other gastrointestinal pathologies such as colorectal cancer and a single bacterial species [30]. Rather, a general community shift in bacterial population has been suggested to contribute to colorectal cancer [31]. However, further work is required to determine if a specific microbial profile can fulfill Koch’s postulate criteria, proving a causative relationship between a colonic microbial profile and colorectal cancer [32]. Within this review, we discuss prominent examples of bacterial species involved in colorectal cancer. An extensive list of pathobionts and symbionts involved in colorectal cancer are summarized in Table 1.

## 3. Gut Microbiota and Colorectal Cancer

Cancer occurrence in the large intestines is approximately 12-fold higher than the small intestines [33]. This may be credited to the much greater bacterial density in the large intestines (≈10^12^ cells per mL) compared to small intestines (≈10^2^ cells per mL) [34]. Supporting a role of microbiota and colorectal cancer, mice that are genetically susceptible to colorectal cancer display a significantly reduced frequency of oncogenic mutations and tumor formation when germ-free, compared to those in the presence of a conventional microbiota [35]. This suggests that the presence of colonic bacteria may potentially contribute to the initiation and progression of colorectal cancer [36]. Similarly, common risk factors for the development of colorectal cancer such as inflammatory bowel disease, obesity, and a diet rich in fat and protein have all been associated with alterations of the microbiota [2,37]. This implies a change in bacterial populations within the gut of colorectal cancer patients, which may include an increase in pathogenic cancer-associated bacteria and a loss of protective anti-cancer bacteria. Supporting this, comparisons of gut microbiota between normal individual and colorectal cancer patients reveals that the predominant flora in colorectal cancer are pathogenic bacteria such as *Escherichia coli* strains harboring polyketide synthase (pks) pathogenicity islands, enterotoxigenic *Bacteroides fragilis,* and *Fusobacterium nucleatum* [38,39,40]. On the other hand, beneficial strains of bacteria such as butyrate-producing species have been shown to be under-represented in colorectal cancers, which is consistent with a reduced amount butyrate being seen in the stool of colorectal cancer patients compared to normal individuals [41]. The relationship between bacteria and colorectal cancer is complex and unclear, with symbionts and pathobionts having conflicting roles in the pathogenesis of the disease [30]. What is also unclear is defining the role of the bacteria in the stages of colorectal carcinogenesis, with bacteria involved in the initiation of colorectal cancer being described as bacterial drivers, which are then gradually outcompeted by opportunistic passenger bacteria that have a competitive advantage within the defined tumor microenvironment. These passenger bacteria can either promote or hinder cancer progression, depending on whether pathobionts or symbionts flourish in the tumor environment. Hence, external factors such as diet and medication have the potential to influence the tumor microbiota, depending on whether they support symbiont or pathobiont bacterial growth [34].

### 3.1. Pathobionts and Colorectal Cancer

The microbiota have the potential to contribute to both the initiation and progression of colorectal cancer through multiple mechanisms including inducing oncogenic signaling, producing carcinogenic metabolites and toxins, and modulating colonic inflammation, which is summarized in Figure 1. Persistent activation of the immune system, as seen in inflammation, can contribute to cancer through the production of growth factors and cytokines that can facilitate tumor growth, perturb differentiation, and promote cancer cell survival [42,43].

Species of the *Clostridium* genus, including *Clostridium perfringens*, *Clostridium hylemonae*, *Clostridium sordelli*, *Clostridium scindens*, and *Clostridium hiranonis*, along with *Bacteroides fragilis*, *Bacteroides vulgatus* and *Listeria monocytogenes*, have been shown to be involved in a multistep deconjugation and biotransformation process that synthesizes secondary bile acids from primary bile acids [44,45,46,47,48,49]. The usual role of primary bile acids is in lipid digestion and cholesterol metabolism; however, they are also implemented in host–microbe interactions. Primary bile acids are usually reabsorbed through the enterohepatic circulation; however, they can also act as substrates in bacterial biotransformation in the colon into secondary bile acids [50]. The secondary bile acids lithocholic acid and deoxycholic acid have been associated with colonic polyps and colorectal cancer through increasing oxidative stress-induced colonic inflammation and activating the oncogenic signaling pathway Wnt [50,51,52,53,54,55]. A study by Nagengast et al. supported the role of secondary bile acids in the initiation of colorectal cancer. They determined that patients with colonic adenomas showed an increase in deoxycholic acids compared to healthy controls [56].

Hydrogen sulfide-producing bacteria such as *Bilophila wadsworthia*, *Bacteroides fragilis,*
*Helicobacter pylori,*
*Clostridium septicum,* and *Streptococcus bovis* have also been implemented in the initiation of colorectal cancer. Hydrogen sulfide is a genotoxic substance that causes genomic instability through damaging DNA [57,58]. Similarly, hydrogen sulfide can diffuse into colonocytes and interfere with mitochondrial function, causing an increase in proliferative signaling through activation of the MAP kinase pathway [59]. Supporting the role of hydrogen sulfide-producing bacteria in colorectal cancer, a study compared the microbiota of African Americans with and without colorectal cancer. A greater abundance of sulfidogenic bacteria in colorectal cancer patients were found when compared to healthy patients [57].

Microbiota associated with the progression of colorectal cancer includes the presence of the pathogenic bacteria *Fusobacterium nucleatum* [60]. *Fusobacterium nucleatum* adheres to the colonic epithelium, where it can invade and induce inflammatory and oncogenic signaling pathways contributing to the growth of colorectal cancer cells. This is induced through *Fusobacterium* adhesin A (FadA)-induced activation of β-catenin signaling, with FadA levels in colonic tissue from patients with adenomas and adenocarcinomas being between 10 to 100 times greater than healthy individuals [61]. This has been shown to contribute to the progression of colorectal cancer, as patients with a high abundance of *Fusobacterium nucleatum* tend to have more advanced disease with a poorer prognosis and shorter survival time [62,63].

*Escherichia coli* is a commensal bacteria residing within the colon, however, pathogenic strains of phylogenetic group B2 harbor the *pks* genomic island that can produce the genotoxin colibactin [64]. Colibactin can induce DNA double-strand breaks and chromosomal aberrations that can contribute to the development of sporadic colorectal cancer [65]. Colibactin-producing *Escherichia coli* has been found in 55–67% of colorectal cancer patients, compared to 20% of control patients [65]. *Escherichia coli* harboring *pks* has been shown to contribute to disease severity in mice predisposed to colorectal cancer [66], supporting a passenger role for *Escherichia coli* in colorectal cancer. However, given the potent genotoxic effects of colibactin and *Escherichia coli* harboring pks being able to induce colorectal cancer in animal models, a driver role is also plausible [67].

Enterotoxigenic *Bacteroides fragilis* is a pathogenic bacterium that produces the enterotoxin *Bacteroides fragilis toxin* (*bft)* and is associated with colorectal cancer initiation and progression [68]. This occurs through modulation of the mucosal immune system and inducing alterations in epithelial cells, leading to a compromised colonic barrier [57,69]. Enterotoxic *Bacteroides fragilis* has been associated with colonic pre-neoplastic lesions and has been suggested to be a potential biomarker for early detection of colorectal carcinogenesis [70]. In a study by Boleij et al., 72.7% of early-stage colorectal tumors had a *bft* gene present, while 100% of late-stage tumors had a *bft* gene present [68]. This supports the role of enterotoxigenic *Bacteroides fragilis* in the progression of colorectal cancer, as late-stage tumors have more carcinogenic bacterial toxin Along with this, the abundance of enterotoxigenic *Bacteroides fragilis* in colorectal mucosa has been shown to be an independent predictor of three-year survival [69,71]. This supports the potential role of enterotoxic *Bacteroides fragilis* as both a driver and pathogenic passenger bacterium in colorectal carcinogenesis [34].

### 3.2. Symbionts and Colorectal Cancer

In contrast, other bacterial populations can have a protective effect against colorectal cancer. For instance, *Bifidobacterium thermophilum* binds free iron to its surface, reducing iron for pathogenic bacterial growth and reducing free radical formation, both of which can contribute to cancer progression [75]. Likewise, *Lactobacillus acidophilus* and *Bifidobacterium longum* have been seen to have a protective effect against cancer, forming a barrier against colonization by pathogenic bacteria and inactivating carcinogenic compounds [98,99].

Gut bacterial populations such as *Bifidobacterium* and Firmicutes produce the short-chain fatty acid (SCFA) butyrate, which is believed to possess anti-cancerous properties. Butyrate is a key energy source for colonocytes, maintaining intestinal epithelium integrity and playing a central role in regulating the stability of the microbiota [100,101,102]. Within colorectal cancer cells, butyrate is able to inhibit histone deacetylase, which leads to increased expression of genes involved in inhibiting the cell cycle and inducing apoptosis [103]. Butyrate can also control cancer through regulating immune homeostasis, leading to a reduction in pro-inflammatory immune cells and cytokines [104]. As butyrate is able to have a protective effect against cancer, this explains why patients with colorectal cancer have significantly reduced abundance of butyrate-producing bacteria compared to healthy patients [100]. Furthermore, murine studies have shown that supplementation with butyrate-producing bacteria and dietary fiber was associated with a reduction in tumor growth [105,106].

A study by Dai et al. assessed bacterial populations that tend to be upregulated and depleted in colorectal cancer. They found five protective bacteria that were decreased in colorectal cancer patients compared to controls. These included *Clostridium butyricum*, which promotes apoptosis of colorectal cancer cells and inhibits intestinal tumor development in mice [59]. This supports the depletion of protective bacterial species that can prevent or limit cancer growth in colorectal cancer, which can potentially be replaced by pathogenic species that support tumors [107].

## 4. Dysbiosis and Bacterial Iron Utilization

Environment, lifestyle, and dietary factors all play key roles in regulating the composition and function of the human microbiota [108]. Hence, these certain factors may alter the microbiota to potentially support tumorigenesis. Obesity, smoking, alcohol, and red and processed meat consumption have all been suggested to have a relationship between the microbiota and colorectal cancer [109]. Smoking has been shown to reduce the abundance of butyrate-producing *Bifidobacterium*, which has anti-inflammatory and anti-tumoral properties [110]. Likewise, obesity has been shown to promote colorectal cancer development through increasing microbial derived pro-inflammatory molecules, such as lipopolysaccharides, along with microbially induced epigenetic alterations [109,111]. Many dietary components have been shown to contribute to colorectal cancer. Red meat possesses procarciongenic properties through increasing secondary bile acids and hydrogen sulfide that can contribute to oxidative stress and cellular proliferation in colorectal cancer. Along with this, dietary heme iron from red meat has been shown to increase mucin-degrading bacteria, such as *Akkermansia muciniphila,* which can lead to an impairment of gut barrier function and contribute to colorectal disease [109].

Colonic nutrient availability is a key regulator of gut microbial populations and is regulated through diet [17]. Many dietary nutrients can regulate colonic bacterial populations, with a major contributor being iron [112]. Iron availability is vital for humans and microbes; hence, both multi- and unicellular organisms have developed strategies to obtain iron from their proximate environment through evolutionarily conserved methods [9]. Iron has a universal role required for protein and enzymatic function, as well as for energy production, and is essential to fundamental biological processes of cell growth and differentiation [113]. Humans obtain iron from their diet and through iron recycling within the body. Dietary iron comes in the form of heme iron from red meat and non-heme iron from dark green leafy vegetables [114,115]. Dependent on iron consumption, around 15% of dietary iron is absorbed within the duodenum, with the remnant passing into the large intestine where it has the potential to be utilized by colonic bacteria. These bacteria are able to produce high-affinity iron-chelating molecules called siderophores, as well as transferrin/lactoferrin receptors, which allow bacteria to scavenge for free iron within the colon. Iron is essential for both the survival and replication of nearly all bacteria [6,9].

A study by Parmanand et al. used an in vitro model to assess the effect of a decrease in gut iron availability on the microbiota. They found that the growth of potentially pathogenic bacterial species, such as *Salmonella typhimurium* and *Escherichia coli,* was significantly inhibited when cultured in an iron-deficient medium, whereas probiotic bacterial species such as *Lactobacillus rhamnosus* were unaffected by iron depletion [116]. This suggests a role of luminal iron in promoting the growth of pathogenic bacteria, but not of probiotic bacteria. This has been assessed in anemic African children who were given iron-fortified biscuits. Iron fortification increased the abundance of pathogenic *enterobacteria* while decreasing the abundance of beneficial *lactobacilli*. This change is also associated with increased gut inflammation, as shown by a raised fecal calprotectin concentration. This indicates that increasing dietary iron within this cohort contributes to gut dysbiosis, as well as suggesting that increasing pathogenic and decreasing beneficial bacterial populations has the potential to contribute to disease through inducing gut inflammation [117].

Iron regulation of bacterial populations within the gut presents a complex relationship, with iron having direct and indirect outcomes on bacterial populations. This can be seen in species of the *Bacteroides* genus that require both heme and non-heme iron for growth [118]. *Bacteroides* species present a prominent proportion of the gut microbiota, with different species showing pathogenic, probiotic, or both properties. Therefore, increasing iron colonic concentration has the potential to directly influence both pathogenic and probiotic *Bacteroides* species [119]. Iron regulation of probiotic *Bacteroides* species involved in the metabolism of undigested dietary fiber can also have an indirect influence on potentially pathogenic strains of *Escherichia coli.* This is through the production of a nutrient niche from the release of sialic acid from mucus and undigested carbohydrates. This facilitates *Escherichia coli* growth as they do not produce their own sialidase enzyme and are reliant on other residential bacteria [120]. Therefore, an increase in iron availability can have an indirect effect on bacterial populations through producing a nutrient niche by probiotic bacteria, which can be utilized to facilitate potential pathogenic bacterial growth, such as *Escherichia coli* harboring *pks* genomic islands [11].

## 5. Iron Supplementation, Microbiota, and Colorectal Cancer

Alterations in the colonic microbiota play a key role in the pathogenesis of colorectal cancer through an increase in pathogenic bacterial populations at the expense of protective probiotic species [121]. Many pathogenic bacteria have heightened iron acquisition mechanisms to aid in their growth and virulence. This can alter microbial populations when there is an increase in gut luminal iron concentration [122,123,124]. Along with diet, oral iron supplementation to treat anemia can also contribute to luminal iron concentration and therefore has the potential to alter colonic bacterial populations [125]. Patients with colorectal cancer often present with iron deficiency anemia, which is commonly induced through chronic tumor-induced blood loss, impairment of iron homeostasis through chronic inflammatory disease, and reduced iron absorption [126]. Anemia in colorectal cancer patients is associated with postoperative complications and poorer patient outcomes; therefore, iron therapy is essential in order to correct anemia peri-operatively [126,127,128]. However, depending upon the route of administration, iron therapy can lead to an increase in luminal iron available for pathogenic bacteria [36]. Oral iron is currently the most common treatment for anemia in colorectal cancer patients, however, along with multiple gastrointestinal side effects such as abdominal pain, dyspepsia, and diarrhea, oral iron has the potential to increase procarciongenic bacterial populations [129,130].

The contribution of iron supplementation to microbial alterations in colorectal cancer has been assessed in murine studies. Constante et al. compared the effect of oral iron supplements to systemic iron supplementation. They found that oral heme iron altered microbial populations, inducing dysbiosis and specifically decreasing butyrate-producing taxa, which ultimately was associated with a decrease in fecal butyrate levels. Thus, oral iron has the potential to worsen disease by reducing butyrate, which has anti-inflammatory and anti-cancerous properties. This was confirmed by demonstrating that dietary heme worsened colitis with greater development of adenoma formation in the mouse model compared to systemic iron [131]. This suggests that the use of intravenous iron supplementation could be more beneficial than oral iron, potentially reducing microbial changes, inflammation, and colorectal cancer progression. These murine studies require detailed exploration with a view to translation into human clinical trials in order to confirm the microbial implications of oral iron therapy before this can lead to a change in the clinical administration of iron to treat anemia in colorectal cancer.

Patients with inflammatory bowel disease are at a higher risk of developing colorectal cancer compared to the general population. Similar to colorectal cancer, the pathology of inflammatory bowel disease involves alterations in the microbiota and is also associated with anemia, with patients requiring iron therapy [132,133]. Human clinical studies investigating oral iron against intravenous iron therapy on the intestinal microbiota in patients with inflammatory bowel disease have been conducted by Lee et al. They found that oral iron differentially altered both bacterial phylotypes and fecal metabolites compared to intravenous iron therapy notably leading to a decreased *Faecalibacterium prausnitzii* and *Ruminococcus bromi* abundance following oral iron therapy [122]. *Faecalibacterium prausnitzii* is an abundant bacteria in a healthy gut microbiota that possesses anti-inflammatory properties through producing butyrate [134,135]. Likewise, *Ruminococcus bromi* is involved in butyrate production through degrading resistant starch particles [136]. A decreased abundance of these bacteria in inflammatory bowel disease patients can, therefore, contribute to the pathogenesis of disease through contributing to inflammation [137]. As inflammatory bowel disease can be a premalignant condition, the microbial mechanisms underpinning the pathology of the condition may have a similarity to those in colorectal cancer [132]. Hence, the negative impact of oral iron therapy on the microbiota in inflammatory bowel disease may act as an indicator of the potential outcomes in colorectal cancer. However, as colorectal cancer can develop from a non-inflammatory bowel disease origin and because the microbial mechanisms are not identical, clinical studies investigating this phenomenon in colorectal cancer are required to fully unravel the potential adverse outcomes of oral iron therapy [138].

## 6. Molecular Pathological Epidemiology in Colorectal Cancer

Along with iron, many other nutritional factors are associated with colorectal cancer, with studies on iron and microbiota having the potential to be confounded by other dietary components. This suggests the need for widescale studies to investigate the interrelationships between environmental exposure such as diet and nutrition, microbiota, and genetics. These factors can be addressed in relation to microbial pathologies and clinical outcomes in colorectal cancer through molecular pathological epidemiology (MPE) [139]. Modifications of the microbiota may not only be a cause of neoplasia but may act as an informative biomarker. Hence, MPE can be utilized to identify probable biomarkers to indicate the potential outcomes of nutritional exposures, such as dietary patterns and medications, on disease outcomes. Along with this, unraveling the complex interplay between diet, microbe, and host may provide the potential for therapeutic intervention through dietary alteration and probiotic bacterial supplementation [140].

Initial MPE studies have investigated the complex relationships between diet, microbiota, and colorectal cancer, assessing potentially pathogenic bacteria such as *Fusobacterium nucleatum, Escherichia coli,* and *Bacteroides fragilis* and probiotic bacteria genera such as *Bifidobacterium* and *Lactobacillus*. A prudent diet pattern rich in fish, poultry, fruit, vegetables, and whole grains has been shown to be associated with a lower risk for *Fusobacterium nucleatum-*positive colorectal cancer, but not *Fusobacterium nucleatum*-negative cancer [141], whereas an inflammatory dietary pattern rich in red and processed meat, sugar, and refined grains is linked with a higher risk of *Fusobacterium nucleatum*-positive colorectal cancers, but not with a risk in *Fusobacterium nucleatum*-negative cancers [142], suggesting that a prudent diet can decrease the risk of colorectal cancer, as well as the fact that nutritional status can regulate the microbiota in order to support colorectal cancer [140]. Likewise, long-term consumption of a diet rich in red meat and fats have been shown to increase the proportion of *Bacteroides fragilis* and *Escherichia coli* within the gut microbiota. An increase in these potentially pathogenic bacteria in response to red meat and high-fat diet leads to an increased risk of colorectal cancer, through contributing to barrier dysfunction, inducing inflammation, and promoting carcinogenic pathways [143]. A study by Kellingray et al. assessed the outcomes of diet on the abundance of sulfate-reducing bacteria. They found that a diet rich in *Brassica* vegetables, such as cabbages, kale, and cauliflower, was associated with a reduction in hydrogen sulfide-producing bacteria. The authors suggest that a diet rich in *Brassica* vegetables can potentially be beneficial to gastrointestinal health through limiting the production of hydrogen sulfide, which can contribute to a reduction in the incidence and progression of cancer [144]. *Bifidobacterium* and *Lactobacillus* are both probiotic genera of bacteria that possess anti-cancer properties through the production of SCFAs. Both genera of bacteria are increased in response to a high-fiber diet and can act to reduce inflammation and lower the risk of colorectal cancer [143,145]. As these bacteria have potential anti-cancer properties, they have been suggested as a potential probiotic therapy following surgery for colorectal cancer. This has been assessed in a study by Zaharuddin et al., which involved a probiotic consisting of six viable *Bifidobacterium* and *Lactobacillus* bacteria given orally twice daily for 6 months. The probiotics were able to modify the intestinal microbiota, leading to a reduction in systemic proinflammatory cytokines following treatment [146]. Future MPE research in the context of nutrition and microbiota is required to investigate the potential outcomes of iron, both dietary and supplementary, on bacterial populations in colorectal cancer, and the potential this has to alter clinical outcomes in these patients.

## 7. Conclusions

The bacteria that make up a healthy colonic microbiota act to maintain homeostasis and should elicit a tolerogenic response. However, dysbiosis can cause an increase in pathogenic bacteria at the expense of protective bacteria, which can contribute to both colitis and cancer. Pathobionts can contribute to disease through compromising the intestinal wall integrity, inducing inflammation, contributing to oncogenic signaling, and producing carcinogenic products such as hydrogen sulfide, secondary bile acids, and bacterial toxins, whereas symbionts can limit inflammation and inhibit tumor cell growth through the production of butyrate, inactivating carcinogenic compounds, and binding free iron to their surface to limit pathogenic bacterial growth. Nearly all bacteria require iron for growth and survival, however, many pathogenic bacteria have highly specialized iron-acquiring mechanisms that aid in their virulence. This has been supported in studies showing that higher luminal iron concentration leads to a greater abundance of pathogenic bacteria, while a decrease in gut iron hinders pathogenic bacterial growth and favors protective bacterial species. As colonic luminal iron concentration may induce alterations in the composition of the microbiota, any factor that will influence iron intake needs to be controlled in patients with pre-existing dysbiosis, such as in patients with colorectal cancer. As patients with colorectal cancer often present with iron deficiency anemia, it may be more prudent to replenish iron stores parenterally rather than enterally so as to not contribute to the luminal iron concentration available for pathogenic bacteria. As the passenger bacterial population in colorectal cancer can either be protective or pathogenic, increased iron availability may support the induction of pathogenic bacteria that may support tumor progression over protective passenger bacteria that may hinder disease progression. This has been supported in murine studies showing that oral iron induces colonic microbial dysbiosis and exacerbates intestinal disease. For this to be translated into the clinical context, clinical trials investigating iron therapy on microbial populations in colorectal cancer patients are required, as the potential microbial outcomes and disease progression associated with iron supplementation may lead to an alteration in the administration route of iron therapy.

## Figures and Tables

**Figure 1 nutrients-12-02512-f001:**
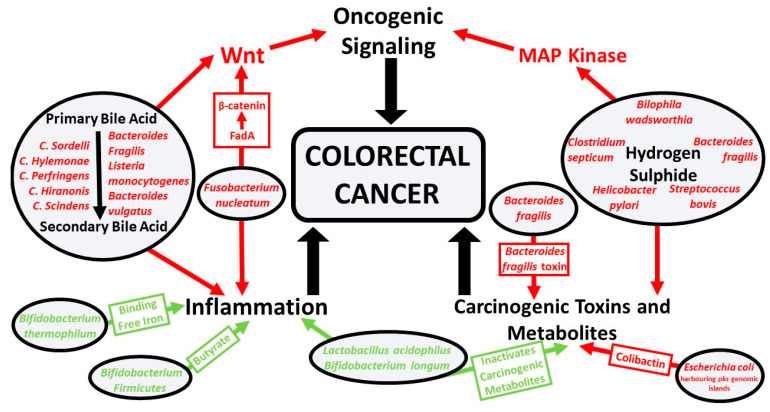
Protective and pathogenic bacterial species and microbial pathways involved in colorectal cancer. The three major microbially induced pathways that contribute to colorectal cancer are inflammation, oncogenic signaling, and carcinogenic toxin and metabolite production. Symbionts (green) inhibit these pathways and pathobionts (red) promote these pathways.

**Table 1 nutrients-12-02512-t001:** Potential mechanisms of symbionts and pathobionts in colorectal cancer pathogenesis.

Phylum	Class	Order	Family	Genus	Species	Symbiont or Pathobiont	Potential Mechanism
Actinobacteria	Actinobacteria	Bifidobacteriales	*Bifidobacteriaceae*	*Bifidobacterium*	*Bifidobacterium bifidum*	Symbiont	Produces metabolites that can inhibit colorectal cancer cell growth [72].
					*Bifidobacterium lactis*	Symbiont	Inhibits NF-κB signaling, limiting colitis-associated colorectal cancer [73].
					*Bifidobacterium longum*	Symbiont	Lactic acid-producing bacteria that inhibits colorectal tumor cell proliferation through modulation of MAP kinase oncogenic pathway [74].
					*Bifidobacterium thermophilum*	Symbiont	Binds free iron, reducing iron for pathogenic bacteria and reactive oxygen species production [75].
		Propionibacteriales	*Propionibacteriaceae*	*Propionibacterium*	*Propionibacterium freudenreichii*	Symbiont	Exerts a protective effect against colorectal cancer through the production of short-chain fatty acids, acetate, and propionate [76].
Bacteroidetes	Bacteroidia	Bacteroidales	*Bacteroidaceae*	*Bacteroides*	Enterotoxigenic *Bacteroides fragilis*	Pathobiont	Degrades E-cadherin and activates beta-catenin signaling, upregulating c-Myc expression and contributing to colonic cellular proliferation [77]. Secretes *B. fragilis* toxin that can induce STAT3 signaling [78].
					*Bacteroides vulgatus*	Pathobiont	Involved in the hydrolysis of primary bile acids to secondary bile acids, which is associated with colitis and oncogenic signaling [49,79]. Correlated with systemic inflammation and colorectal cancer tumor stage [79].
			*Rikenellaceae*	*Alistipes*	*Alistipes finegoldii*	Pathobiont	Linked to colitis-associated colorectal cancer; in vivo studies have shown to be through activation of IL-6/STAT3 signaling [80].
Firmicutes	Bacilli	Lactobacillales	*Enterococcaceae*	*Enterococcus*	*Enterococcus faecalis*	Pathobiont	Can produce reactive oxygen and nitrogen species that can contribute to DNA damage and colonic inflammation [81].
			*Lactobacillaceae*	*Lactobacillus*	*Lactobacillus acidophilus*	Symbiont	Can produce conjugated linoleic acids from linoleic acid. Fatty acids produced by these species act on colonocytes, possessing antiproliferative and proapoptotic mechanisms [82].
					*Lactobacillus casei*		
					*Lactobacillus delbrueckii*		
					*Lactobacillus plantarum*		
			*Streptococcaceae*	*Streptococcus*	*Streptococcus bovis/gallolyticus*	Pathobiont	Can induce production of cytokines that lead to free radical production, colonic inflammation, and increased angiogenesis. Contributes to pro-proliferative signaling via MAP kinase and COX-2/prostaglandin induced cellular proliferation and inhibited apoptosis [83].
		Bacillales	*Bacillaceae*	*Bacillus*	*Bacillus subtilis*	Symbiont	Probiotic bacteria that can inhibit the proliferation of colorectal cancer cells, induces cell cycle arrest, and promotes apoptosis. Shown to reduce inflammation and aids in immune homeostasis [59].
	Clostridia	Clostridiales	*Clostridiaceae*	*Clostridium*	*Clostridium butyricum*	Symbiont	Produces butyrate, which possesses anticancer properties, inducing cell differentiation and apoptosis, as well as inhibiting cellular proliferation [84,85].
			*Lachnospiraceae*	*Eubacterium*	*Eubacterium rectale*	Symbiont	Anti-inflammatory butyrate-producing bacteria [86].
				*Roseburia*	*Roseburia intestinalis*		
			*Peptostreptococcaceae*	*Peptostreptococcus*	*Peptostreptococcus anaerobius*	Pathobiont	Has been shown to interact with Toll-like receptors 2 and 4 on colonic cells. Elevating levels of reactive oxygen species, promoting cell proliferation and increasing colonic dysplasia [87].
			*Ruminococcaceae*	*Faecalibacterium*	*Faecalibacterium prausnitzii*	Symbiont	Butyrate-producing bacteria, shown to be anti-inflammatory and able to inhibit colorectal tumorigenesis [88,89,90].
Fusobacteria	Fusobacteriia	Fusobacteriales	*Fusobacteriaceae*	*Fusobacterium*	*Fusobacterium nucleatum*	Pathobiont	FadA medicated activation of beta-catenin can contribute to inflammatory and oncogenic signaling [61]. Fap2 protein present on *F. nucleatum* binds to TIGIT on the antitumor immune cells, natural killer cells, and T-cells, which can limit tumor immunosurveillance [91].
Proteobacteria	Deltaproteobacteria	Desulfovibrionales	*Desulfovibrionaceae*	*Bilophila*	*Bilophila wadsworthia*	Pathobiont	Proinflammatory sulfate-reducing bacteria capable of producing genotoxic hydrogen sulfide [92].
	Gammaproteobacteria	Enterobacterales	*Enterobacteriaceae*	*Escherichia*	*Escherichia coli (*harboring *pks* pathogenicity islands)	Pathobiont	Produces the bacterial genotoxin colibactin that promotes the growth of colonic tumor cells. Colibactin induces DNA interstrand crosslinks and double-strand breaks [93,94].
				*Salmonella*	*Salmonella typhimurium*	Pathobiont	Produces the bacterial protein AvrA, which is associated with inflammation and colorectal cancer, through modulation of the p53 pathway [95,96].
Verrucomicrobia	Verrucomicrobiae	Verrucomicrobiales	*Akkermansiaceae*	*Akkermansia*	*Akkermansia muciniphila*	Pathobiont	A mucin-degrading bacterium that contributes to colonic inflammation [97].

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
