# Peer review of "Influence of Iron on the Gut Microbiota in Colorectal Cancer"

_nutrients, 2020, doi:10.3390/nu12092512_

Round 1

Reviewer 1 Report

This is a generally well-written review on the roles of iron and microbiota in carcinogenesis. The topic is of very high interest. This paper can be improved after taking following points.

Many discussion points are binary and dichotomous (eg, bacteria positive vs. negative; cell growth vs. no growth). It is a bit misleading because many biological characteristics form or represent continuum. 

There are many environmental, dietary, and lifestyle factors (beyond iron) that influence the microbiome, immune system, carcinogenic mechanisms, and response to therapy. Actually data on iron may be confounded by other components in human studies. The authors should discuss these points; influence of those factors, eg, smoking, diet, obesity, etc.

There are also influences of germline and somatic genetic variations on both oncogenic signaling and immune function/microbiota. Those can be discussed. Currently discussion is only one-way (microbiome influencing tissue and tumor).

In these contexts, as a future direction, research on dietary / lifestyle / genetic factors, microbiome, immunity, and molecular tissue biomarkers is needed. The authors should discuss molecular pathological epidemiology (MPE), which can investigate those factors in relation to molecular pathologies, immunity, and clinical outcomes in cancer. MPE, its strengths and challenges have been discussed in Annu Rev Pathol 2019, J Pathol 2019, etc. I believe MPE research can be a promising direction.

Reviewer 2 Report

The aim of this review is to detail the influence of iron on the gut microbiota in colorectal cancer (CRC). Instead it focuses largely on the reported pertubation of gut microbiota (dysbiosis) in patients with this disease and how this can lead to an increase in the relative abundance of “pathogenic” bacteria at the expense of protective species. While the suggestion is made that pathogenic bacteria have heightened iron acquisition systems and that oral iron therapy given to CRC patients who present with anaemia is likely to drive dysbiosis of the gut microbiota, this hypothesis is not well developed.

Specific comments

The abstract suggests that to limit CRC disease progression external factors need to be tightly controlled in patients with CRC but there is no consideration given to how control of these external factors, such dietary iron, might also limit the risk of bacteria-associated pre-cancerous changes, which is surely just as important.

Section 2.2 “Bacteria and cancer” focuses on H. pylori-associated risk of gastric cancer. It is unclear why this section then goes on to detail the potential mechanisms underlying the reported association between H. pylori infection and increased risk of iron deficiency anaemia. There is no mention of the potential influence of iron on the outcome of long-term colonisation with these bacteria, given that this is the overarching theme of the review

Section 3 covers gut microbiota and colorectal cancer, and suggests that the presence of colonic bacteria may potentially contribute to the initiation and progression of CRC. However, the statement that “pathogenic bacteria such as B. fragilis and E. coli are associated with an increased risk of cancer” is incorrect. The authors cite Lin et al (Ref 34) but that paper clearly describes that it is carriage of toxigenic strains of B. fragilis and E. coli that are implicated in increased risk of disease, not B. fragilis and E. coli per se. Moreover, with regards E. coli, it is carriage of strains that carry the pks pathogenicity island and express the colibactin toxin that is associated with increased risk of CRC, as opposed to other toxins that may be elaborated by unrelated strains of E. coli. This difference, which is extremely important, is not recognised by the authors either in the text or in Table 1, and may well apply to many of the other bacterial species designated as “Pathogenic”. Are are all these strains considered pathogenic, or are some more likely to be “pathobionts” because they have the potential to express virulence factors?

The authors suggest that most pathogenic bacteria have heightened iron acquisition methods and therefore tend to outcompete protective bacteria for free iron but is there any actual evidence for this? For example, Bacteroides spp. require heme iron for growth. There are approximately 19 species of Bacteroides and a range of these found in the gut microbiota in most individuals, where they are largely considered beneficial for the human colon because of their role in the breakdown of undigested dietary fibre. Moreover, their ability to release sialic acid from mucus and undigested carbohydrates creates a nutrient niche not only for themselves, but also for E. coli that do not secrete a sialidase enzyme and must reply on bacteria like Bacteroides spp. to colonise the mucus layer. The authors state that microbiome analysis has identified over-representation of colonic mucosa-associated E.coli in patients with colorectal cancer. Is it possible that this might reflect an indirect effect, where heme-mediated growth of Bacteroides spp. facilitates cross feeding of microbiota such as E. coli (known as the “restaurant” hypothesis).

Changes in the gut microbiota in patients with colorectal cancer are widely documented. However, it is unclear whether the dysbiosis reported in these individuals is a cause or a consequence of the disease. i.e. does a changing environment lead to the increased relative abundance of certain bacterial species. If so, would limiting external influences such as diet, prevent further shift in these microbial populations and thereby limit carcinogenesis?

Considerable emphasis is given to the idea that under normal physiological conditions the commensal bacteria act to outcompete the potential pathogenic colonization of the colon, therefore preventing infection. This seems to be based on the assumption that gut microbiota associated with CRC are acquired as the result of infection. More likely (and using H. pylori as an example) an individual’s gut microbiota is acquired at an early age and it is increasingly considered likely that long-term colonisation with toxin=producing strains of gut bacteria underlies an individual’s risk of developing this disease. In this scenario, bacteria that use dietary iron are likely to flourish and, if these bacteria are toxigenic, then increased growth is likely to equate to increased toxin production and, downstream, increased risk of disease.

The authors refer to Harold Tjlsma’s excellent review, where he describes how some bacterial species are more likely drivers of colon carcinogenesis whereas other species are more likely to be passengers with regards their role in the genesis of CRC. This is something the authors might like to reflect on.

Minor comments

Figure 1 is difficult to interpret. For example, the Bacteroides fragilis toxin is not a carcinogenic metabolite. Also, how do Lactobacillus acidphilus and Bifidobacterium longum inactivate BFT and/or DNA damage?

Table 1 details an extensive list of “pathogenic bacterial species” and the potential mechanisms that link these bacteria to CRC pathogenesis. Section 3.1 details protumorigenic colonic microbiota, some of which are not detailed in Table 1 (and vice versa). Does this imply that the mechanisms underlying protumorigenic differ from those of pathogenic bacteria?

Round 2

Reviewer 1 Report

The authors successfully addressed the points and improved the paper. 

Author Response

We would like to thank the reviewer for their input and expertise in improving our manuscript. 

Reviewer 2 Report

This review is largely about the effect of diet on the gut microbiota. i.e. the title is not in line with most of the content.

Line 205 suggests ETBF are associated with CRC progression through modulation of the mucosal immune system and inducing alterations in epithelial cells leading to a compromised colonic barrier. However, there is also evidence of a role for ETBF in the development of pre-neoplastic lesions (Purcell et al), an observation that fits with Tjlasma's hypothesis that long-term colonic carriage of bacteria such as ETBF has the potential to initiate and drive carcinogenesis. i.e  these changes likely precede mutation of the APC that, in turn, increases the risk of adenoma formation. 

The revised figure has removed reference to DNA damage and yet this is the proposed pathway for the link between carriage of pks+ve strains of E. coli and increased risk of CRC. While the potential for an iron-mediated increase growth of E. coli is discussed, but there is no mention how this might increase risk of disease of individuals colonised with the potentially genotoxic strains of E. coli. Consider including discussion of this in section 3.1.
